# Utilisation of Digital Applications for Personal Recovery Amongst Youth with Mental Health Concerns

**DOI:** 10.3390/ijerph192416818

**Published:** 2022-12-14

**Authors:** Vicki C. Dallinger, Govind Krishnamoorthy, Carol du Plessis, Arun Pillai-Sasidharan, Alice Ayres, Lillian Waters, Yasmin Groom, Olivia Alston, Linda Anderson, Lorelle Burton

**Affiliations:** 1School of Psychology and Wellbeing, University of Southern Queensland (UniSQ), Darling Heights, QLD 4350, Australia; 2Child and Youth Mental Health, Children’s Health Queensland, Queensland Health, Queensland Government, Brisbane, QLD 4000, Australia

**Keywords:** personal recovery, recovery-oriented care, recovery colleges, youth, digital mental health interventions

## Abstract

There is an increasing population of youths that report mental health issues. Research has shown that youths are reluctant to seek help for various reasons. A majority of those who do seek help are using digital mental health supports. Subsequently, efforts to promote youth mental health have focused on the use of digital applications as a means of overcoming barriers related to factors including stigma and lack of available services. The worldwide move toward recovery-oriented care led to emerging research on personal recovery amongst youths with mental health concerns. This study sought to address the need for recovery-oriented digital resources for youths. It utilised a qualitative design methodology to develop a rich interpretation of how youths are using digital interventions to support their mental health recovery journey. It sought to understand how existing digital applications are useful for youth recovery and identified characteristics associated with recovery and engagement. The content analysis generated five categories that represent facilitators of youth recovery and the thematic analysis identified key elements of digital applications that support youth recovery. The results offer complimentary support and guidance for recovery-oriented care and the use of digital mental health interventions in the promotion of personal recovery amongst youths.

## 1. Introduction

Problems with mental health (MH) among 15–24-year-olds are growing globally [1,2,3]. Up to 20% of youths have MH concerns [3]. Seventy-five percent of MH problems begin before the age of 18 [4]. MH disorders disrupt development, place burdens on families, and influence adulthood health and social functioning [5,6,7]. Providing at-risk youths with professional MH care is a significant concern [8]. Youths that require MH support are least likely to seek it [9,10]. For example, 75% of UK children with MH illnesses are not receiving treatment [11]. Systematic reviews of barriers to youth MH help-seeking identified stigma and negative attitudes to MH to be the most common [12,13]. Stigma is the fear of being socially sanctioned or embarrassed, inhibiting certain activities, such as help-seeking and under-reporting MH concerns [14]. It also embodies the negative social judgement and attitudes toward MH and MH treatment [15]. Martínez-Hernáez et al. [14], in their systematic review, found that half of the included research alluded to stigma and stigmatising views towards MH conditions as barriers to help-seeking. Stigmatising beliefs leads to fears of vilification by others (“they’ll think I’m crazy”) and self-stigmatisation (“why can’t I get over this myself” [14]).

Stigmatising beliefs in youths include perceptions of MH services as impersonal, protocol- and diagnosis-driven, and indifferent to individual needs [14]. Youths perceive MH professionals as reducing their problems to psychiatric diagnoses. Additionally, youths find pathophysiological explanations for MH concerns challenging to understand and anxiety-inducing [13]. Biomedical attributions of MH are linked to public judgments of those with MH concerns as impulsive, unpredictable, and hazardous [16,17]. Biomedical conditions are also often viewed as chronic and unchangeable. According to the optimal distinctiveness theory [18], diagnosing and labelling reduces within-group differences and exaggerates between-group disparities. This promotes MH stigmas [19]. Biomedical conceptualisations of mental illness overemphasise the person over their social surroundings [17]. Clinical discourses on MH focus on risk factors, issues, and deficiencies [20]. Such discourses focus on the negative features of those MH difficulties, reinforce stigmatising beliefs, and poorly represent the protective factors, strengths and capacities of persons with MH concerns [16].

Personal recovery is an alternative, stigma-reducing paradigm for conceptualising MH concerns [20,21]. Personal recovery is focused on the idiosyncratic journey of growth and development, supporting individuals to function to the best of their ability with or without symptoms or issues associated with their MH [20,22,23]. Personal recovery is different from ‘clinical recovery,’ which aims to reduce psychiatric symptoms and restore function [20]. Leamy and colleagues [21] highlighted the following five basic components of personal recovery: connectedness, hope, identity, purpose, and empowerment (referred to in the acronym CHIME). Figure 1 depicts the five framework components. The CHIME model of recovery has a growing body of data to support its value in influencing clinical and research programs and is regarded as the model of choice for defining and exploring recovery in the field of MH [24,25,26,27].

The notion of recovery for youths is relatively new [28]. While research shows relevance to younger populations, there are additional developmental considerations for this population, including the involvement of caregivers and systemic supports [2,6,29,30,31,32,33,34,35]. Dallinger et al. [35] conducted a multisystemic qualitative analysis with youths, caregivers, MH professionals and educators to conceptualise youth recovery. They identified that while CHIME processes are relevant to youth recovery, developmental considerations require these processes to operate on a recovery continuum. They established that youth recovery requires a unique balance between restoration and resilience polarisations of each of the CHIME recovery processes (see Figure 2). Restoration processes follow a more paternalistic and protective course, with a focus on rebuilding and acceptance; while resilience processes provide a path forward developmentally with new connections, skills, and a focus on strengths.

In addition to the value of the recovery model for youth, it is also clear that youth have other preferences in relation to how they access MH support. Youth often express high levels of mistrust for MH professionals regarding privacy and patient confidentiality [13,14]. Youth report fears that privately disclosed information in therapy may be communicated to their families [14]. Given these concerns about privacy and lack of anonymity, youth often turn to the internet for assistance with their MH [36]. Research has shown that youth prefer to access MH resources online [37,38,39,40]. Approximately 22.2% of 13–17-year-olds use online services for help or information about emotional or behavioural problems [41]. In addition to anonymity, the internet and digital application have several advantages, including minimisation of risks of contracting COVID-19; no geographical boundaries or limitations to the access of resources and services being free or at low cost to the user [42,43]. Online mental health services can provide interactive solutions to engage young people in a self-directed and anonymous way, thus assisting and supporting overburdened face-to-face services [8,37,39,43]. Dallinger et al. [44] conducted a systematic review of internet-based mental health programs for youth to assess recovery-oriented elements of the interventions and outcomes. While several programs reviewed included features aligned with CHIME recovery processes, no digital intervention explicitly targeted youth recovery. In addition, several online programs reviewed identified concerns with early drop-out and poor adherence to treatment protocols [45,46,47,48,49,50,51,52,53,54]. Given the growing number of digital applications, websites and resources aimed at youth, Dallinger et al. [44] highlighted the need to develop a broad understanding of how youth utilise existing digital applications to support their recovery needs. 

Recovery is a relatively new concept when applied to youth and research has demonstrated the application of recovery to youth is unique and multidimensional, not unlike their help-seeking behaviours. The increasing prevalence of MH issues in youth calls for new approaches to MH delivery. However, several barriers have been identified in youth accessing traditional face-to-face services including fear of stigma, cost, confidentiality, and reduced availability [36,55,56]. DMH platforms offer a familiar and trusted medium for youth to access and explore the concept of recovery and related interventions in ways that tailor to their individual needs. DMH platforms also offer the potential to in-crease the reach and engagement of our younger population with these much-needed recovery resources. Building on previous research, the present study aims to understand youth perspectives on how digital applications are supporting restorative and resilience focused recovery processes and what are the specific characteristics and functions of these applications. The intent of the study was to explore how youth are using digital applications within the youth recovery journey.

## 2. Materials and Methods

The data were derived from workshop discussions and interviews with youths across Queensland, Australia, who had or were experiencing MH issues and receiving either outpatient psychotherapy or inpatient treatment within a medical treatment facility. Ethics approval was obtained through the University of Southern Queensland Human Research Ethics Committee (H20REA100) as part of a larger program of research. Participant contributions throughout the focus groups and interviews were anonymised during the transcription process. Semi-structured interview methods were used to explore the use of digital recovery resources. Data were combined to provide a holistic view of the use of digital recovery applications.

### 2.1. Participants

Participants were recruited through the following two treatment modalities: a public medical MH facility and a private practice (see Figure 3). Participant details are noted in Table 1.

Participation was voluntary, all participants completed a consent form and both participants and caregivers were provided with written information detailing the nature of the study. Participants under the age of 18 years were also required to provide caregiver/guardian consent. Participants and caregivers were informed that focus groups and interviews would be audio/video recorded before participation. Participants were provided a copy of interview questions and research project information before the interview to allow for reflection. Participant inclusion criteria were that participants had to be aged between 15 and 24 years (youth as defined by United Nations [57]); and have a diagnosed MH dis-order. Participants considered by their treating health professionals to be experiencing severe symptoms were excluded from participating. A purposive sample of 17 youths was recruited through local public and private MH services and consumer advocacy groups. Data was collected between 11th February 2021 and 29th April 2022. Participants were eligible to receive retail vouchers valued at AUD$100. Participants were offered the option of interview or focus group data collection.

### 2.2. Procedure

Interviews were used to explore a variety of views regarding youth recovery and how key recovery program elements and priority areas for recovery education would translate to a digital recovery platform for young people. Characteristics of digital resources that promote recovery and engagement were also explored. Questions were broad and not inclusive of the word recovery to allow for novel themes to emerge. All interviews were audio-recorded, and field notes were taken. Data collection included two rounds of interviews and one focus group.

The interviews were conducted across a variety of healthcare settings across Queensland. Some interviews were conducted over the phone due to COVID-19 restrictions at the time of data collection. The interviews ranged between 20 and 60 min in duration. Participants were asked to answer four questions developed by the research team (see Appendix A). The focus group was held at a state-run youth public health service facility in Queensland, Australia, and ran for 90 min in duration with two participants (originally, 4 consented, although 2 declined to participate on the day). Focus groups were facilitated by a researcher and consumer program facilitator at Queensland Health. Focus group participants were asked to answer three questions developed by the research team that related to feedback on informational recovery video prototypes presented in the focus group and interviews (see Appendix B). Questions focused on the use of digital resources in supporting youth MH. The researcher allowed time for introductions and provided a brief outline of the research for all interviews and the focus group. An opportunity to debrief and provide additional thoughts was offered after each session.

### 2.3. Data Analysis

The focus group and interview material were analysed using a qualitative deductive and inductive directed content analysis to explore youth digital lived experience perspectives and their relevance to youth MH recovery processes [58,59]. Content analysis is a systematic procedure that codes and analyses qualitative data, and a combination of deductive and inductive approaches can be used for multiple approaches to analysis [60]. This analysis included a three-step approach which was as follows: deductive, inductive, and summary. Deductive content analysis is appropriate when an existing theory in-volves the application of conceptual categories in the analysis of a novel perspective. Furthermore, data that does not correspond to an existing theory is ascribed to new codes and categories and follows an inductive approach. For instance, the data about digital activity were initially tallied and then analysed inductively using Hsieh and Shannon’s [58] four-step qualitative data analysis strategies and ascribed to the existing recovery processes (CHIME, restorative, and resilience). Initially all transcripts were read for data immersion and to obtain a comprehensive understanding of the data as a whole. The recordings were listened to several times, while transcribed data and field notes were read, re-read, and checked against recordings for accuracy. Codes were identified from the data that included text representative of key concepts. Thirdly, initial coding was applied to all data and the potential for new codes was accommodated. Finally, codes were analysed into categories. This data was then viewed through the lens of recovery processes. Deductive analysis was applied by ascribing digital resource application types to each relevant recovery process (connectedness, hope, identity, meaning, and empowerment) within an existing youth recovery model (restorative and resilience). Lastly, the summary seeks to find the relationship between both deductive and inductive analyses. 

Deductive thematic analysis was used to explore the key characteristics of digital re-sources through the lens of youth MH and recovery as applied to the Model of Internet Intervention [61]. This model draws on previous theory and identifies the relationships among the various components to support symptom improvement and behaviour change within a digital context. Thematic analysis followed the six steps outlined by Braun & Clarke [62].

To increase rigor, the coding of data was conducted by four researchers separately. All researchers utilised NVivo software. Each researcher worked individually to encode all information and produce categories and interpretations. The coding of data and categories was then reviewed collectively in an iterative process until consensus was met between researchers on key categories and their related sub-categories.

## 3. Results

### 3.1. Sample Characteristics

The sample population consisted of 16 youths and 62.5% identified as female, with a mean age of 16.5 years. The youths presented with various mental health diagnoses (see frequency Table 2). The mean number of days in treatment was 227 days.

### 3.2. Inductive Analysis

Using the CHIME processes as a framework enabled the investigation of digital resources and their application to youth recovery. The inductive content analysis provided a tally of digital applications and websites used by this youth population with MH issues (see Table 3). These tallies provided a synopsis of digital use within the research population and demonstrated a range of experiences and reasons for use and their potential relationship to recovery processes.

Contextual information from the interviews provided a framework for the relationship between the experience and purpose of use and existing CHIME processes. Youths discussed their purpose for using the various applications and clinical reasoning was also applied when allocating relevance to CHIME processes. For example, youths spoke of using YouTube as a way of connecting with the world and journaling was used to reflect and build a sense of hope and empowerment through achievement. Journalling is also empirically associated with identity formation [63]. The tally of digital resources and applications relevant to CHIME processes is provided in Table 4.

### 3.3. Deductive Analysis

The deductive analysis was used to view the use of digital resources through existing recovery processes (CHIME; [21]) and their alignment to existing youth recovery processes of restoration and resilience [35]. The remaining data formed a new category and related sub-categories: key elements of digital applications supporting youth recovery).

#### 3.3.1. Connectedness

The deductive analysis offered several sub-categories related to connectedness which are presented in Table 5. Youth spoke of using digital resources to help them with communicating with family and friends about their MH concerns. Youth reported using digital resources to share difficult experiences they struggled to describe. Using digital mediums offered the space and support that youth sometimes need to feel comfortable engaging with close connections after periods of stress and dysregulation. Often youth are challenged to understand what is happening within the recovery journey and feel more confident using information sourced digitally to explain their symptoms and needs to others. They used messaging apps for brief communications and to keep in contact with family and friends. It was also important for youth to have immediate access to caregivers, family, and peers to support them throughout their recovery. Youth used applications to share updates and information on health and MH status—including in times of crisis. There was consensus around using digital applications to share pictures and videos from the past to reconnect with family and friends.

Some youths spoke of reading and participating in MH forums and shared interests to connect with MH communities. Participating in online gaming platforms was identified as a way to communicate with friends and meet new people. They also discussed meeting new people through digital applications and online communities. Participant discourse highlighted using websites that include a crisis or support webchat function and crisis telephone lines to link to supports and/or engage in telehealth consultations with health and welfare professionals. Some youth referred to using alternative forms of communication such as web chat as beneficial to accessing professional connections and support. They noted that the idea of speaking face to face is overwhelming and deters them from help-seeking or seeking connection with supports. When accessing new connections and community supports, youth reported digital resources as the preferred medium, as they offer flexibility with anonymity and selecting connections. For some, having alternative means of connection was described as lifesaving during times of need. All youths spoke of using phones and video chat to stay connected to family and friends. Some youth reported using digital resources as a way of lightening the mood and creating a more relaxed environment, thereby initiating repair to damaged relationships caused during episodes of illness.

#### 3.3.2. Hope

Table 6 presents the sub-categories related to the category of hope. Youths spoke of using websites and applications (including webchats and online questionnaires) to source information about MH symptoms and diagnoses to increase MH literacy. Participant discourse highlighted the use of MH websites to develop an understanding and prognosis for diagnoses and symptom management. Digital applications were utilised to share progress and celebrate gains regarding symptoms, maladaptive behaviours, and mood progress towards goals that support MH. Participants discussed using social forums, web chat, and phone calls to experience hope through connection and information in times of need. Youths spoke of using digital resources to monitor and track the progress of goals and other focuses, such as tracking the frequency of self-harming behaviours.

For participants, digital resources offered a confidential and accessible medium for storing goals and benchmarks that are easily accessible and under their supervision, rather than the supervision of family or professionals. Hope is manifested through the tracking of personal goals and the privacy afforded by digital mediums that offer a space for youths to develop insight into their MH and coping strategies. Participant discourse highlighted the need to have access to connections when they are most needed. Several youths spoke of help-seeking during the night or at times outside of typical business hours.

The use of social media and lived experience websites was reported as a way for youths to learn from those who have overcome difficult times and succeeded in their goals, despite MH challenges. There was a consensus that peer-lived experience stories offered understanding and normalisation. By listening and watching others speak of their struggles and achievements throughout their recovery journey, youths found hope through revaluating expectations of themselves and others within the context of their MH and life in general. Most participants referred to accessing these narratives through digital media, as they felt this offered more authentic and honest stories. Narratives also helped to improve their understanding of the recovery journey and its impact on themselves and their families, redefining expectations and supporting feelings of hope in the process. Normalising this process provided hope and optimism within their current and future circumstances. Youths reported using apps to set and persevere with goals and benchmarks related to MH and the future. Videos and search engines were used to access positive, humorous, and uplifting content to remind youths that they can be happy. Youths spoke of using digital resources as a form of delight and distraction and noted that they often use them to engage family and friends in uplifting experiences. Participants spoke of using applications and websites to make new friends and engage in developmentally normative conversations. Youths identified using digital resources to foster a sense of community and interact with existing and new connections to support their normative development.

#### 3.3.3. Identity

The deductive analysis provided several sub-categories related to recovery process identity and these are presented with exemplar quotations in Table 7. Youths found that looking at pictures and videos from the past allowed them to reconnect with neglected parts of their identity and sense of self. They reported that sharing information online, including interests that they have in common with family and friends, was significant to their sense of self and identity and affirmed their connection. Youth discourse highlighted the use of social media to maintain normative relationships and identification with family, friends, and community groups. By sourcing and identifying with similar recovery stories, youths noted a better understanding of their MH, enabling them to distinguish themselves from their symptomology and MH experiences.

Youths spoke of using websites to explore the prevalence of MH concerns within peer groups and finding identification with these populations. They mentioned listening to stories and experiences of post-traumatic growth as part of a positive identity related to MH concerns. Most participants spoke of using digital resources to support normative developmental processes by engaging with peers in social forums and sharing digital experiences. Through exploring online communities related to sub-cultures and interests not associated with MH concerns, they reported that digital resources offer an escape from the focus on MH and allow them to be themselves. This included social forums, games, social media, and websites. Knowing others are also experiencing the journey was important for youths to understand themselves better. Understanding their experiences and MH offers the opportunity for normalisation and affiliation. It is important in developing a healthy sense of identity and was part of positive identity development related to MH concerns.

#### 3.3.4. Meaning

Table 8 presents the sub-categories related to the category of meaning. Moving on from one’s past was essential to youth recovery and finding meaning within their individual experiences. Participants spoke of the need to share their experiences and loss of opportunities resulting from their mental illness with like-minded peers. Sharing this grief with others who are experiencing similar issues was important for youths to find a purpose behind what has happened. Youths spoke of using digital forums and social media as channels for these experiences. They discussed exploring MH websites to build an understanding of expected challenges, losses, and grief during the recovery journey and using social media and digital devices to reflect and grieve through photos and records of past experiences.

Youths also reported that digital forums and social media were preferred to find meaning in shared experiences. Participants said that while they sometimes spoke of the loss of opportunities, they also found meaning through their connections with others in similar situations. Affiliations with lived experience peer groups and role models offered youths the opportunity to view their own stories as meaningful and worthy of sharing. Through engagement with MH professionals using webchat, forums, websites, and telehealth, youths discussed the acquisition of new understandings of MH conditions, challenges, and experiences. Participants reported using digital resources to source credible and accurate information to make meaning of their symptoms and experiences with MH. Youths spoke of using applications to document and record moments in their recovery journey and communicate the emotional impact of MH.

#### 3.3.5. Empowerment

The deductive analysis provided several sub-categories related to the category empowerment and these are presented with exemplar quotations in Table 9. Digital resources such as social media provided youths access to lived experience narratives and coping strategies. These narratives offered an improved understanding of MH experiences and the need for support, adjusting youth perspectives around receiving help. Youths reported that often, there is a need to accept support to build capacity and coping skills. Particularly for adolescents, accepting support was described as a complex process. While support can be helpful at times, it can also be disempowering if provided in a controlling, coercive, and paternalistic manner. Youth discourse reflected that journalling and tracking achievements and experiences promoted a sense of self-efficacy and self-agency within their recovery journey.

Several youths mentioned using the internet and applications as a distraction to support coping and emotion regulation. Several participants identified the benefits of peer stories of lived experience, MH, and recovery. Youths found connecting to peers who have had similar experiences valuable to their recovery. Some youths reported finding new affiliations with peers or groups that represented this lived experience population. Participants spoke of using digital resources to learn new information and coping strategies. Youths felt a sense of control over their recovery when accessing these tools through digital means at any time and any location. Youths highlighted the benefits of instant access to anonymous and confidential support in times of greatest need. Youths also spoke of using social media and forums to share stories of MH lived experiences and offering support to others in this way. Several youths spoke of using the internet and apps for fun and positive mood promotion and feeling empowered by sharing these experiences with significant others in their life.

#### 3.3.6. Key Elements of Digital Applications Supporting Youth Recovery

Participants discussed the characteristics of useful and engaging digital applications. Table 10 provides a summary of the themes associated with these elements. It provides an interactive method to maintain an organic context and allows participants to engage in content development.

Most participants preferred the use of bright and calming colours. Youths spoke of the benefits of personalised tracking and goal-setting functions and their benefits related to motivation and MH. Many participants noted that engagement with digital applications depended on whether the resources were concise or easy to navigate and understand. The inclusion of a help function and feedback channel was necessary for participant recovery. It was important for all participants to access educational content for recovery, social connections and forums, lived experience narratives, caregiver resources, and evidence-based interventions for use at home. Participants recommended the inclusion of videos or animated content. Youths emphasised several elements involved in the source and style of the content that supported engagement and impact for youth. These included less emphasis on MH, making it more relevant to what is happening in the world now, using strength-based content and ensuring it is authentic. They also discussed the value of co-design, interaction, choice, fun, inspiration, and gratification. Participants also noted the need to include measures of recovery and symptoms to personalise the experience and increase engagement.

### 3.4. Summary

Both analyses demonstrated the relationship between digital resources and youth recovery. Strength in the processes of connectedness, hope, and empowerment was evident across both deductive and inductive analyses. Deductive analysis showed that youths use digital applications to support connectedness and empowerment, while the inductive analysis also indicated that they find hope in using digital applications. Evidence for identity and meaning was also present, although less prominent within the results.

## 4. Discussion

The purpose of the present study was to investigate how digital applications can support youth recovery. It also sought to explore how digital applications and their characteristics would translate to youth recovery interventions and promote engagement with youth recovery digital resources. Three core themes developed from the data highlighted the relationship between recovery and the use of digital tools supporting youth. Firstly, it is important to reflect on these results in the context of the broader findings on how youth use digital supports and research on help-seeking behaviour. Previous research has shown that youth spend a significant amount of time on the internet connecting with others and searching for information on MH and that when they do seek support this is often the medium they chose to use [36,39,42]. With some research showing that digital interaction can lead to exacerbation and development of MH issues [64,65] it is important that the development of future digital resources work to support MH and mitigate these challenges where possible. The Model of Internet Interventions offers such a framework [61]. This research has shown that youth utilise digital applications in a way that aligns with the youth recovery framework of restoration and resilience processes proposed by Dal-linger et al. [35] and subsequently demonstrates the application of CHIME factors through these two processes.

Youth use digital resources for connection, research, distraction, and relaxation to manage their symptoms. They seek to hear the stories of others with MH journeys and to connect to other youth who may be experiencing similar challenges, helping them feel validated and connected to the community. Digital applications provide access to re-sources such as music, guided meditation, and links to services supporting MH. They are used to track symptoms and progress, as reminders, and for instant gratification and connection to alter mood states or provide a sense of self-worth. 

Youth are using digital resources and DMH applications to support recovery. They are connecting and building connections through social media and the sharing of digital resources that promote both restorative and resilience processes. Youth prefer this mode of communication to face-to-face interaction because it affords a sense of anonymity and confidentiality that is absent from more traditional MH interventions and services. Having the opportunity to listen to stories of lived experiences from peers around the country and the world offers an understanding of their own experience providing a sense of hope for youth that more clinical interventions rarely achieve. Youth with MH concerns find a sense of affiliation and identification through these narratives and connections. Digital platforms and applications also offer youth the opportunity to engage in normative activities with peers and family, supporting developmentally appropriate connections and al-lowing youth to identify with their peer group. Connections and information available through digital resources provide an opportunity for youth to find meaning in their experiences and to establish a sense of purpose through validation and recognition of their journey. Accessing tools and coping strategies through this means creates a sense of self-agency and empowerment. 

The processes of hope, identity, and meaning seemed to have less resonance within the data from this study. While identity-building processes may be less evident within the use of digital resources, there is still an opportunity for digital applications to act as a medium for this process. It is also indicated within this study that digital resources alone may not offer significant meaning-making or hope-building opportunities. However, there is a window of digital opportunity that is available for therapeutic supports and ROP. Perhaps in conjunction with therapeutic support digital resources would offer youth a more directed and scaffolded approach to meaning-making and identity formation. A blended therapeutic and digital experience provides a level of scaffolding that is both developmentally appropriate and targeted in the engagement of youth. Through digital pre-scription and professional support on digital platforms there lies the potential for a digital ecosystem supporting both self-directed recovery and in-person therapy.

The conceptualisation of youth recovery and ROP for youth is still emerging within the field of recovery research. This research has offered support for the integration of digital and DMH applications within both youth recovery and recovery interventions for youth. Additional support may benefit recovery processes of identity and meaning through targeted intervention and digital prescription within a therapeutic context. There is a need for youth MH services to move toward ROP and recovery-focused support. Efficacy for the recovery college suggests this as an appropriate service to offer conjunct with other MH services and therapeutic resources in support of this transition. The concept of an online recovery college for youth offers the potential to include several relevant digital resources that have been vetted for authenticity and accuracy; providing a one-stop location for youth to learn and explore their recovery with confidence and security. This re-search has shown that youth are already accessing recovery-oriented interventions and that an online recovery college may offer a more central and focused resource for youth recovery. 

There were several key elements of how youth identified and engaged with digital applications that aligned with the Behavioural Model of Internet Interventions [61]. Participants identified several appropriate and relevant elements of digital applications that they felt would translate to a recovery-oriented application. These elements align with Ritterband et al.’s [61] Behavioural Model of Internet Interventions and may guide the development of an online recovery college and consequently provide a template for future research in this area.

### 4.1. Study Limitations

Limitations to this study included the number and location of participants, with many participants coming from metropolitan areas within Queensland. Th original scope of re-search was to include narratives of youth across multiple settings. This research has pro-vided youth perspectives from inpatient and private settings across metropolitan and rural locations; however, this population is not exhaustive. These findings, while relevant, are to be applied with caution as there are several limiting factors to their generalisability. They are indicative of youths that are aware of and have access to, private and public health services. As Australia is a high-resource country, they do not offer a global representation of the socio-economic impacts of recovery-oriented care and perspectives. Having the viewpoints of other youth around Australia and worldwide would add to the validity of this research. Finally, they do not include adequate representation of marginalised populations and Aboriginal and Torres Strait Islander youth. 

Other limitations exist within the chosen method of analysis. Content analysis offers a method of interpretation and should be applied with relative caution. It is often evaluated for trustworthiness. This research was based on the narratives of 17 youth, purposively sampled to ensure results were dependable and relative to youth MH and recovery. Content saturation was achieved within these narratives. While data was analysed and consensus sought between more than one researcher, multiple interpretations of this data would exist. By providing a description of the analytical procedures used and presenting quotations from the narratives, we hoped to demonstrate trustworthiness of the analysis and reliable findings. All of these limitations highlight the need for further recovery re-search among youth populations.

### 4.2. Implications for Practice and Future Research

This study demonstrates a need for additional research that investigates the application of digital recovery-oriented practice applied to younger populations. While this work and other research have shown that recovery models can be applied to younger populations, there is more work that must be carried out in the examination of elements such as language and the role of caregivers within the recovery journey. Additionally, understanding how to better access and support this population through digital platforms holds promise for greater reach and engagement. Additional research is also required to add to these data and increase the understanding of how youths prefer to seek help and engage with recovery-oriented digital supports. While the need for online recovery support for youths has been identified through this research, the barriers and risks must also be considered when inviting a more vulnerable population such as youths to engage in DMH supports. Future research in the monitoring and vetting of DMH resources to reduce the potential for exposure to maladaptive coping skills, predators, and breaches in confidentiality and anonymity is required.

## 5. Conclusions

This research is the first of its kind to offer a series of digital characteristics that rein-force digital engagement for youth in support of recovery. While several studies have evaluated the efficacy of digital mental health programs [66,67,68], there was limited re-search on how youth utilised digital applications to promote personal recovery and self-management of mental health concerns. The findings of this study provide key insights for future digital youth recovery resource development, an area that is in dire need of further study. Understanding youth digital engagement offers valuable details for sup-porting help-seeking and promoting recovery interventions within the digital ecosystem. Most youths use digital platforms every day and many are seeking help for MH issues. This research has provided an opportunity to explore how best to support youth seeking help through digital means. It offers a novel opportunity to explore how to develop evidence-based supports for youth recovery using their preferred medium. The present study has provided several key elements of recovery-oriented digital care for youth. It has demonstrated how youth currently use digital resources for recovery and identified how digital supports promote recovery for youth. By exploring the use of digital resources this research has contributed to the understanding of youth recovery processes. It delivers opportunity for digital prescription within clinical intervention and treatment programs that promotes engagement and self-agency within the youth recovery journey. In conclusion, it offers support for the development of ROP and targeted recovery interventions and a pre-liminary framework for the development of an online youth recovery college.

## Figures and Tables

**Figure 1 ijerph-19-16818-f001:**
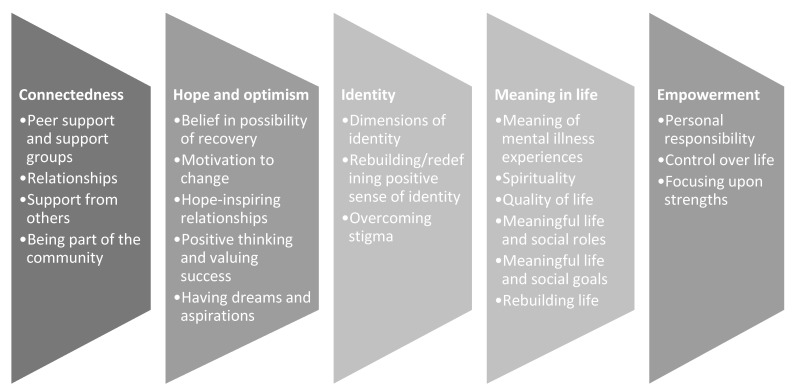
Processes of adult recovery (CHIME; Leamy et al., 2011 [21]).

**Figure 2 ijerph-19-16818-f002:**
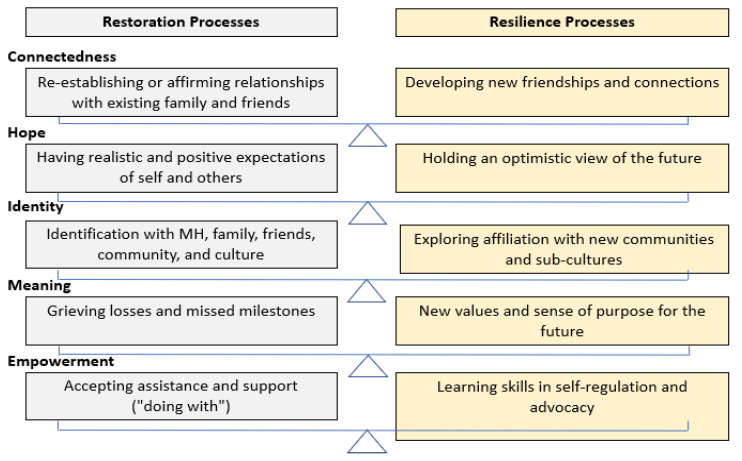
Restoration and resilience processes for youth recovery.

**Figure 3 ijerph-19-16818-f003:**
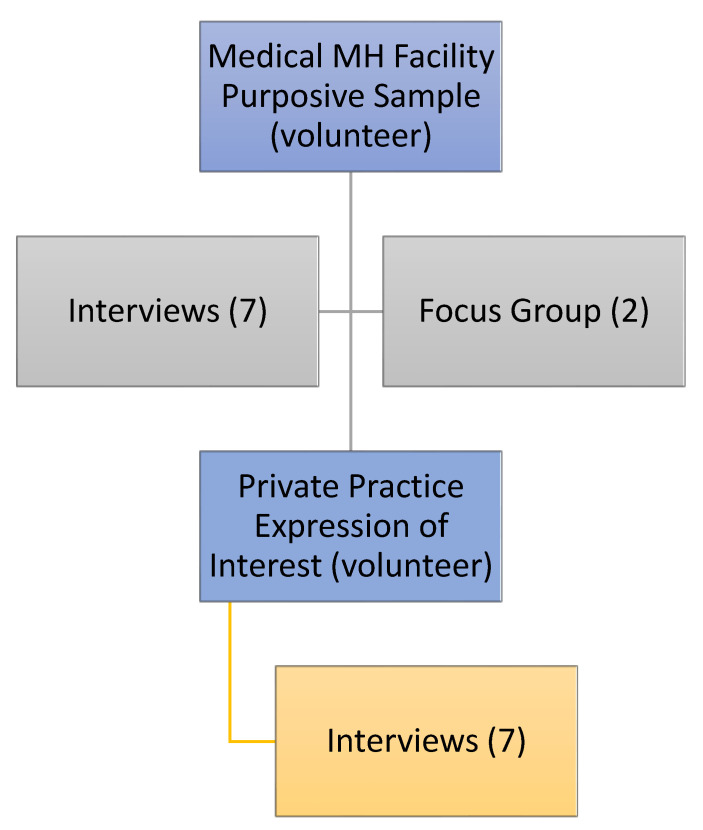
Flow chart of participant recruitment.

**Table 1 ijerph-19-16818-t001:** Participant demographic information.

Gender	Age	Residence	Diagnosis	Days in Treatment
Female	15	Metropolitan	Social phobias	538
Male	15	Metropolitan	Other reactions to severe stress	625
Non-binary	15	Metropolitan	Dissociative motor disorder	128
Female	16	Regional	OCD	243
Non-binary	16	Metropolitan	PTSD	173
Female	16	Metropolitan	Eating disorder	244
Trans female	16	Remote	Depression	121
Female	16	Metropolitan	Conversion disorder	106
Female	16	Rural	GAD, eating disorder	132
Male	16	Metropolitan	ARFID	237
Female	16	Metropolitan	ADHD, GAD	316
Female	17	Metropolitan	OCD	167
Female	17	Rural	ASD, ADHD, PTSD	238
Female *	18	Metropolitan	Trauma and stressor-related diorders	98
Female	18	Rural	GAD	267
Male *	21	Metropolitan	Depression, psychosis	0 **

Note: * = focus group participants; ** = not currently in treatment. Days in treatment are sourced through professional health records. OCD = obsessive-compulsive disorder; PTSD = post-traumatic stress disorder; GAD = generalised anxiety disorder; ARFID = avoidant and restrictive food intake disorder.

**Table 2 ijerph-19-16818-t002:** Tally of relevance to CHIME processes.

Diagnosis/MH Symptoms	Frequency
Anxiety	5
Stress related	4
Eating disorder	3
OCD	2
Depression	2
ADHD	2
Somatic	2
ASD	1
Psychosis	1

Note: OCD = obsessive-compulsive disorder; PTSD = post-traumatic stress disorder.

**Table 3 ijerph-19-16818-t003:** Tally of digital resources, purpose of use for youths, and relevance to CHIME.

Tally	Digital Resource	Purpose of Use	CHIME Processes
1	Video streaming (e.g., YouTube)	Distraction	C
3	Social media (e.g., Tik Tok, Instagram, Snapchat)	Connection/distraction	C, I
3	Mindfulness and meditation (e.g., Calm)	Mood regulation	E
5	Music streaming (e.g., Spotify, Apple Music)	Mood regulation/connection	C, I, E
2	Story reading applications (Abide, Audible)	Mood regulation/distraction	C, M
3	MH websites (Beyond Blue; Kids Helpline, Lifeline)	Meaning-making/connection	C, M
4	Journalling applications	Reflection	H, I, E
5	Goal setting/planning (e.g., Forest, Sober, Beyond Now)	Goal setting/planning/self-regulation	M, E
1	Gaming applications	Distraction/connection	C

Note: C = connectedness, H = hope, I = identity, M = meaning; E = empowerment.

**Table 4 ijerph-19-16818-t004:** Tally of relevance to CHIME processes.

Tally	CHIME Processes
15	Connectedness
4	Hope
8	Identity
10	Meaning
17	Empowerment

**Table 5 ijerph-19-16818-t005:** Exemplars and categories related to connectedness recovery needs.

Generic Category	Sub-categories	Exemplars
Restorative Processes	Communicating difficult experiences	*I guess it would be easier to just say download this App and tell my parents to like have a look at this.*
	Brief communications	*Just being able to talk to family and friends and having that interaction.*
	Communicating in times of need	*It was nice just to feel like someone was listening to me when I was messaging the support worker like they connected me to someone.*
	Reconnecting with others through reminiscing	*…and like even just last night, we were sitting with the chairs, and we were playing through all of our favourite songs…*
	Scheduling	*…and then I probably like set up what I think tomorrow is going to be like … and just how I can like do it better I guess.*
Resilience Processes	Connecting to virtual communities	*You know, somewhere where people who have MH can log into and become part of a community.*
	Online gaming communities	*I used to like play games, sometimes with my friends. Just like mini-games I play with my partner when I can’t see them.*
	Chat groups	*Somewhere to just talk about other stuff that you like, not just your issues.*
	Online contact with health & welfare professionals	*Because I didn’t feel comfortable talking on the phone because I didn’t know who it was and I’m not really comfortable talking on the phone anyway. But it was nice just to feel like someone was listening to me when I was messaging the support worker like they connected me to someone. They gave me resources to…*

**Table 6 ijerph-19-16818-t006:** Exemplars and categories related to Hope recovery needs.

Generic Category	Sub-categories	Exemplars
Restorative Processes	Information about MH prognosis	*I’ve been on like websites to learn more about, like my diagnosis and diagnoses and stuff.*
	Sharing progress and celebrating gain	*… it’s like how long you like stop doing these negative like bad coping mechanisms.*
	Being Heard	*You can ring somebody, and it doesn’t matter what time of day it is, somebody is there to chat to or get back to you.*
Resilience Processes	Stories offer encouragement and hope	*I like reading about like other people who have recovered or are learning to get through it as well. Who have experienced similar things and I guess just… strategies, and kind of like things to do because when you are having a bad day, you can’t really see why you are doing it, like why you are trying. So maybe just something to remind you that you do want this.*
	Support for striving	*… like you can take notes and put your thoughts in it. And if you like, relapse or if you are just having a rough day, it gives you things like contact info and like just little quotes and it reminds you of why you’re doing it. It’s cool.*
	Delight and distraction	*I sort of am the DJ of the household and like during cooking groups and stuff, I put the music on.*
	Sharing delight	*I like making people laugh. Yeah, with jokes or little comments.*
	Hope through belonging	*Like your special interests and stuff like that. So, some way to share and to help other people with their stuff.*

**Table 7 ijerph-19-16818-t007:** Exemplars and categories related to identity recovery needs.

Generic Category	Sub-categories	Exemplars
Restorative Processes	Reconnecting with self through reminiscing	*…and like even just last night, we were sitting with the chairs, and we were playing through all of our favourite songs…*
	Affirming and celebrating identity	*It’s a faith-based one so some of it is like a … Biblical like scriptures and they play stories from the Bible.*
Resilience Processes	Information about prevalence of MH concerns	*I kind of like statistics like you know, like how many people have this. Like this type of therapy is best.*
	Stories of positive identities related to MH	*Maybe some information from people who have done it themselves.*
	Building a new sense of self	*Somewhere to just talk about other stuff that you like, not just your issues if that makes sense… like your special interests and stuff like that.*

**Table 8 ijerph-19-16818-t008:** Exemplars and categories related to meaning recovery needs.

Generic Category	Sub-categories	Exemplars
Restorative Processes	Vicarious grieving: Redefining past opportunities	*Maybe some information from, I guess maybe people who have done it themselves.*
	Reflecting on the past	*…and if I’m feeling down or sad, I’ll play music that is familiar to me so that I can, like, feel something familiar or sad. This is familiar to me because I feel that pretty much all the time.*
Resilience Processes	Discovering growth from adversity	*Just like some stories that people have gone through and how they, like did it but also how they’re saying it’s not going to be this easy or it’s not easy but keep trying and stuff.*
	Understanding MH	*They gave me resources to, perhaps refer to for things that I was worried about. Or for further reference or whatever.*
	Making meaning through Expression	*Maybe like a way to keep a journal. Because I feel like that’s very helpful to kind of see patterns.*

**Table 9 ijerph-19-16818-t009:** Exemplars and categories related to empowerment recovery needs.

Generic Category	Sub-categories	Exemplars
Restorative Processes	Building Acceptance	*It’s going on like to kind of give you like example or something like in real life.*
	Sharing progress and celebrating gain	*…maybe like a way to keep a journal. Because I feel like that’s very helpful to kind of see patterns. You know, and what’s happening…*
Resilience Processes	Shifting Attention	*I was listening to music to try and just get me to go like get more creative and stuff. And I use it to focus on my schoolwork and I use it to calm down if I’m having panic attacks and stuff.*
	Using the Voice of Experience	*…where you can talk to people who go through similar things and sort of see what helps them and what helps you and sort of try to help each other.*
	Shopping for Success	*…if I’m having urges to engage in negative behaviours, I try to look at the good coping strategies and replace them.*
	Having a Voice	*I think I personally know that I’ve gone through a lot and the fact that I’m still here and that I could give good and helpful advice to people.*
	Normative Regulation	*I feel like a lot of other young people use Spotify or like Apple music just to calm down like their emotions when they listen to music.*

**Table 10 ijerph-19-16818-t010:** Key elements of digital applications supporting youth recovery generic and sub-categories.

Theme	Descriptions	Exemplars
Appearance	Bright colours and simple layout with large font	*Yeah, brightly coloured. Vibrant, just like a happy feeling.*
Assessment	Personalised goal setting and tracking	*So different customisable features to make it so sort of more personalised.*
Behavioural prescriptions	Easy navigation with help functions and feedback channel	*Sometimes when I find an App it looks complicated, just don’t end up using it because it’s too hard.*
Reducing burdens	A free resource that comprises short interventions in a safe environment supporting confidentiality	*I feel like maybe it would be nice to see some information from like psychs and stuff. I do like to see how other people deal with it.*
Content	Educational interventions including lived experience and forums for connection supporting youth, caregivers, and professionals	*I think like an online portal when there’s a chat 24/7 would be good… probably people that have been there and done that and can offer advice…*
Delivery	Presenting information with talking heads, animations, and text to support all preferences for learning	*I feel like a video would be cool, too. I mean, everyone learns, or everyone takes in information differently, so I feel like there should be variety to make sure everyone can use it and benefit from that information.*
Message	Co-developed content that is catchy and relevant to the present and younger population	*But there’s opportunities at places like <facility> where you can all grow together and learn together.*
Participation	Include inspirational and fun measures of recovery through games and interactive quizzes and trackers	*I feel like I would use it when I kind of wanted to do something to help myself but couldn’t really think what to do. So that whole thing about activities or ideas, or processes. Make yourself feel better, happier.*

## Data Availability

The data analysed during the current study are not publicly available due to client confidentiality but are available from the corresponding author upon reasonable request; however, restrictions apply to the availability of these data.

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
