# Peer review of "Utilisation of Digital Applications for Personal Recovery Amongst Youth with Mental Health Concerns"

_ijerph, 2022, doi:10.3390/ijerph192416818_

Round 1

Reviewer 1 Report

“Mental health” term should include in the title.

Table 1 must include in the result section. Could you present the results of table 1 in a global way? The mean of age and day treatment, and the frequency of each mental illness and residence. 

Could authors indicate the specific treatment added to the digital application which patients received? Medication, psychology...

Why don't you include any questions about the satisfaction of the digital application?

In the methods, authors should include information about which characteristics should have the digital application used by the participant of the study. 

“Some interviews were conducted over the phone”. Could you explain that? Are some people evaluated face to face and others over the phone?

In table 2, I think it is more useful to indicate the data as I propose in table 1. 

The sample size was small, which is a limitation of the study. Could you provide a flow diagram of the selection process of the participants?

Author Response

Thank you for your comments. They are much appreciated and have been incorporated into the paper. 

Reviewer 2 Report

The article is relevant to the area of mental health and presents well-achieved parts.

However, it presents aspects that require reformulation, namely:

- The summary is not clear in relation to the objectives of the study, nor the main results obtained, not explaining the thematic areas and categories found. As the objectives are not clear, the alignment with the conclusions is compromised;

- Material and methods, line 126, where it refers to “Various interview methods were used”, should immediately clarify the type of interviews used in the study or refer its approach to the next point;

- The discussion is partially supported by other studies, following non-uniform referencing standards throughout the work;

- Standardization of the bibliographic reference standard used and revision of the numbering sequence.

Author Response

(The authors gave the same response as above.)

Reviewer 3 Report

This study examined how young people manage their personal recovery by using available digital applications. The issue of mental health is, fortunately, starting to lose its stigma in the real world, and this article is one more important step to achieving the goal of fully shattering the barriers for people to come out and seek help. Therefore, I commend the authors for tackling such an important topic. In my assessment, the paper shows no signs of a flawed theoretical basis and/or methodological deficiencies that could jeopardise the results. Thus, there is no need for further substantial improvement of the manuscript. However, one minor issue still needs to be resolved. That is, this study’s position in the current literature needs to be clarified. In other words, the author(s) needed to devote more attention to clearly disclosing the study’s contributions. A small paragraph should suffice to hammer the point of the study’s relevance. Since this is an easily manageable concern, the article should be published when it is resolved. Well done!

Author Response

(The authors gave the same response as above.)
